# Transfer to a Low-Resource Language via Close Relatives: The Case Study on Faroese

**Vésteinn Snæbjarnarson**[1,2]    **Annika Simonsen**[3]    **Goran Glavaš**[4]    **Ivan Vulić**[5]

[1]University of Copenhagen    [2]Miðeind ehf    [3]University of Iceland
[4]University of Würzburg    [5]University of Cambridge

## Abstract

Multilingual language models have pushed state-of-the-art in cross-lingual NLP transfer. The majority of zero-shot cross-lingual transfer, however, use *one and the same* massively multilingual transformer (e.g., mBERT or XLM-R) to transfer to *all* target languages, irrespective of their typological, etymological, and phylogenetic relations to other languages. In particular, readily available data and models of resource-rich sibling languages are often ignored. In this work, we empirically show, in a case study for Faroese – a low-resource language from a high-resource language family – that by leveraging the phylogenetic information and departing from the 'one-size-fits-all' paradigm, one can improve cross-lingual transfer to low-resource languages. In particular, we leverage abundant resources of other Scandinavian languages (i.e., Danish, Norwegian, Swedish, and Icelandic) for the benefit of Faroese. Our evaluation results show that we can substantially improve the transfer performance to Faroese by exploiting data and models of closely-related high-resource languages. Further, we release a new web corpus of Faroese and Faroese datasets for named entity recognition (NER), semantic text similarity (STS), and new language models trained on all Scandinavian languages.

## 1 Introduction

Massively multilingual Transformer-based language models (MMTs) such as mBERT (Devlin et al., 2019), XLM-RoBERTa (Conneau et al., 2020a) and mT5 (Xue et al., 2021) have been the driving force of modern multilingual NLP, allowing for rapid bootstrapping of language technology for a wide range of low(er)-resource languages by means of (zero-shot or few-shot) cross-lingual transfer from high(er)-resource languages (Lauscher et al., 2020; Hu et al., 2020; Xu and Murray, 2022; Schmidt et al., 2022). Cross-lingual transfer with MMTs is not without drawbacks. MMTs' representation spaces are heavily skewed in favor of high-resource languages, for which they have been exposed to much more data in pretraining (Joshi et al., 2020; Wu and Dredze, 2020); combined with the 'curse of multilinguality' – i.e., limited per-language representation quality stemming from a limited capacity of the model (Conneau et al., 2020a; Pfeiffer et al., 2022) – this leads to lower representational quality for languages underrepresented in MMTs' pretraining. Cross-lingual transfer with MMTs thus fails exactly in settings in which it is needed the most: for low-resource languages with small digital footprint (Zhao et al., 2021). Despite these proven practical limitations, the vast majority of work on cross-lingual transfer still relies on MMTs due to their appealing conceptual generality: in theory, they support transfer between any two languages seen in their pretraining. Such strict reliance on MMTs effectively ignores the linguistic phylogenetics and fails to directly leverage resources of resource-rich languages that are closely related to a target language of interest.

In this work, we attempt to mitigate the above limitations for a particular group of languages, departing from the 'one-size-fits-all' paradigm based on MMTs. We focus on a frequent and realistic setup in which the target language is a low-resource language but from a high-resource language family, i.e., with closely related resource-rich languages. A recent comprehensive evaluation of the languages used in Europe[1] scores languages

---

[1]The *Digital Language Equality in Europe by 2030: Strategic Agenda and Roadmap* published by the European Language Equality Programme (ELE), `https://european-language-equality.eu/agenda/`.

based on the available resources. Languages such as German and Spanish score at around 0.5 of the English scores, and more than half of the languages are scored below 0.02 of the English score. Many, including almost all regional and minority languages such as Faroese, Scottish Gaelic, Occitan, Luxembourgish, Romani languages, Sicilian and Meänkieli have the score of (almost) 0. However, what differentiates these languages from low-resource languages from Africa (e.g., Niger-Congo family) or indigenous languages of Latin America (e.g., Tupian family) is the fact that they typically have *closely related high-resource languages as 'language siblings'*. In this case, we believe, language models (LMs) of closely related high-resource languages promise more effective transfer compared to using MMTs, plagued by the 'curse of multilinguality', as the vehicle of transfer.

In this proof-of-concept case study, we focus on Faroese as the target language and demonstrate the benefits of *linguistically informed* transfer. We take advantage of available data and resources from the closely related but much more 'NLP-developed' other Scandinavian languages.[2] We show that using "Scandinavian" LMs brings substantial gains in downstream transfer to Faroese compared to using XLM-R as a widely used off-the-shelf MMT. The gains are particularly pronounced for the task of semantic text similarity (STS), the only high-level semantic task in our evaluation. We further show that adding a limited-size target-language corpus to LM's pretraining corpora brings further gains in downstream transfer. As another contribution of this work, we collect and release: (1) a corpus of web-scraped monolingual Faroese, (2) multiple LMs suitable for Faroese, including those trained on all five Scandinavian languages, and (3) two new task-specific datasets for Faroese labeled by native speakers: for NER and STS.

## 2 Background and Related Work

**Cross-Lingual Transfer Learning with MMTs and Beyond.** A common approach to cross-lingual transfer learning involves pretrained MMTs (Devlin et al., 2019; Conneau et al., 2020a; Xue et al.,

2021). These models can be further pretrained for specific languages or directly adapted for downstream tasks. A major downside of the MMTs has been dubbed the *curse of multilinguality* (Conneau et al., 2020a), where the model becomes saturated and performance can not be improved further for one language without a sacrifice elsewhere, something which continued pretraining for a given language alleviates (Pfeiffer et al., 2020). Adapter training, such as in (Pfeiffer et al., 2020; Üstün et al., 2022), where small adapter modules are added to pretrained models, has also enabled cost-efficient adaptation of these models. The adapters can then be used to fine-tune for specific languages and tasks without incurring catastrophic forgetting.

Other methods involve translation-based transfer (Hu et al., 2020; Ponti et al., 2021), and transfer from monolingual language models (Artetxe et al., 2020; Gogoulou et al., 2022; Minixhofer et al., 2022). Bilingual lexical induction (BLI) is the method of mapping properties, in particular embeddings, from one language to another via some means such as supervised embedding alignment, unsupervised distribution matching or using an orthogonality constraint (Lample et al., 2018; Søgaard et al., 2018; Patra et al., 2019), and has also been used to build language tools in low-resource languages (Wang et al., 2022).

Attempts to alleviate the abovementioned issues have been made, such as vocabulary extension methods (Pfeiffer et al., 2021), which add missing tokens and their configurations to the embedding matrix. Phylogeny-inspired methods have also been used where adapters have been trained for multiple languages and stacked to align with the language family of the language of interest (Faisal and Anastasopoulos, 2022). Some analysis on the effects of using pretrained MMTs has been done: Fujinuma et al. (2022) conclude that using pretrained MMTs that share script and overlap in the family with the target language is beneficial. However, when adapting the model for a new language, they claim that using as many languages as possible (up to 100) generally yields the best performance.

Inspired by this line of research, in this work, we focus on improving MMT-based cross-lingual transfer for a particular group of languages, those that have sibling languages with more abundant data and resources.

**NLP Resources in Scandinavian Languages.** A fair amount of language resources have been devel-

---

[2]The Scandinavian languages are a family of Indo-European languages that form the North Germanic branch of the Germanic languages. The largest languages of the family are: (1) Danish (population 5.8M), Norwegian (5.4M) and Swedish (10.4M) – the Mainland Scandinavian languages, and (2) Icelandic (373K) and Faroese (54K) – the Insular Scandinavian languages.

oped for the Scandinavian languages, particularly if aggregated across all languages of the family. It is also worth mentioning that Danish, Icelandic, Norwegian and Swedish are represented in raw multilingual corpora such as CC100 (Conneau et al., 2020b) or mC4 (Xue et al., 2021) as well as in parallel datasets such as (Schwenk et al., 2021; Agić and Vulić, 2019). Large multilingual language models have been trained on these datasets (Devlin et al., 2019; Liu et al., 2020; Xue et al., 2021) but have been shown to have limited capacity for languages with smaller relative representation in pretraining corpora. Faroese is not included (at least not correctly labelled) in these crawled corpora.This may be in part due to the limited amount of Faroese that can be found online, and in part due to its close relatedness to the other languages of the Scandinavian family (Haas and Derczynski, 2021). A brief overview of prior work in cross-lingual transfer to Faroese is given in Appendix D.

In this work, we use the following open language resources for the Scandinavian languages.

**Danish:** The Danish Gigaword Corpus (Strømberg-Derczynski et al., 2021) is a billion-word corpus containing a wide variety of text.We also use a NER resource, the DaNE corpus (Hvingelby et al., 2020).

**Icelandic:** With Icelandic as the most closely related language to Faroese, we experiment with an Icelandic language model, IceBERT (Snæbjarnarson et al., 2022). For the NER experiment, we make use of the MIM-GOLD-NER corpus (Ingólfsdóttir et al., 2020).

**Norwegian:** The Norwegian Colossal Corpus (NCC) (Kummervold et al., 2022) contains 49GB of clean Norwegian data from a variety of sources, making it the largest such public collection in the Nordics. We also make use of the NorNE (Jørgensen et al., 2020) NER corpus (both for Bokmål and Nynorsk).

**Swedish:** The Swedish Gigaword Corpus (Eide et al., 2016) contains text from between 1950 and 2015. The latest NER corpus for Swedish is Swe-NERC (Ahrenberg et al., 2020), where the authors include more modern texts than in earlier corpora.

**Faroese:** A POS corpus, the Sosiualurin corpus is an annotated Newspaper corpus with 102k words (Hansen et al., 2004). The Faroese Wikipedia has also been used to create a tree bank (Tyers et al., 2018), which has a Universal Dependencies

(UD) mapping. We use this corpus along with the FarPaHc (Ingason et al., 2012), which also has a UD mapping.

## 3 New Faroese Datasets

### 3.1 Faroese Common Crawl Corpus (FC3)

Faroese monolingual data is scarce, mainly because of the limited size of the Faroese-speaking population. Despite this, we manage to extract a decent amount of varied Faroese text from the Common Crawl corpus (FC3). To this effect, we adopted the approach of Snæbjarnarson et al. (2022) for Icelandic, i.e., we targeted the websites with the Faroese top-level domain (.fo). After clean-up and deduplication, the obtained Faroese corpus consists of 98k paragraphs containing in total 9M word-level tokens. Albeit relatively small compared to corpora from other Scandinavian languages, this Faroese corpus still drives significant downstream performance gains (see §5).

### 3.2 Named Entity Recognition (FoNE)

We annotate the Sosialurin corpus (6,286 lines, 102k words) with named entities following the CoNLL schema using an Icelandic NER-tagger trained using the ScandiBERT model, see §4. The annotation was then manually reviewed. Out of the 118,533 tokens (including punctuation), 9,001 are annotated using the Date (546), Location (1,774), Miscellaneous (332), Money (514), Organization (2,585), Percent (115), Person (2,947) and Time (188) tags. We refer to this new dataset as *FoNE*.

### 3.3 Semantic Similarity (Fo-STS)

The STS Benchmark (Cer et al., 2017) measures semantic text similarity (STS) between pairs of sentences. For each pair of sentences, the annotators assigned the score (on a Likert 1-5 scale) that indicates the extent to which the two sentences are semantically aligned. We manually translated from English to Faroese 729 sentence pairs from the test portion of the STS Benchmark; the translation was carried out by a native speaker of Faroese fluent in English, who was instructed to preserve in the translation the extent of semantic alignment between the original English sentences.

## 4 Model Training

We train the following new language models: (i) *ScandiBERT* is trained on concatenated corpora of all Scandinavian languages, (ii) *ScandiBERT-no-fo*

is trained on concatenated corpora of all Scandinavian languages except Faroese (i.e., without any Faroese data, that is, no FC3, Bible or Sosialurin), and (iii) *DanskBERT* which is trained only on the Danish data; we train *DanskBERT* for the purposes of comparison with IceBERT, in the setup in which we carry out downstream transfer to Faroese by means of a monolingual model of a closely related language (with Danish being more distant to Faroese than Icelandic). We additionally evaluate transfer with models that have been further pretrained on the FC3 corpus (indicated with the *-fc* suffix). We provide an overview of all training datasets and hyperparameter configurations used in our experiments in Appendix A.

## 5 Experiments

### 5.1 Downstream Performance for Faroese

**Experimental Setup.** In addition to the models presented in §4, we make use of the monolingual Icelandic model IceBERT and the massively multilingual XLM-on-RoBERTa (XLM-R).[3] We evaluate the performance of this set of pretrained models in several downstream tasks in Faroese: Part-of-Speech tagging (POS), Dependency Parsing (DP) (UD datasets introduced in §2), Named Entity Recognition (NER), and Semantic Text Similarity (i.e., the new NER and STS datasets introduced in §3). For all downstream tasks the task-specific training and evaluation data span monolingual Faroese data points only: we carry out the experimentation via ten-fold cross-validation on the respective Faroese datasets.[4] For each model and downstream task, we carry out ten runs with different random seeds (each run trains the model for 5 epochs with batches of 16 instances) and report the average performance across runs. The exception is the STS training in which the models were fine-tuned for 3 epochs (with training batches of size 8).[5]

**Results and Discussion.** Table 1 summarizes the

results across the four downstream tasks. The best-performing model for POS, as evaluated on the Sosialurin POS corpus, is ScandiBERT-fc3, outperforming ScandiBERT by more than 1 point in terms of F1. However, the ScandiBERT-no-fo-fc3 model, without any Faroese data at pretraining, obtains fully on-par performance with the variant that does include Faroese data.

The best-performing model for NER, and STS is the ScandiBERT-no-fo-fc3 model. Somewhat surprisingly, we get the best performance for the model that does not include any Faroese data in the initial pretraining, that is, it does not adjust the tokenizer/vocabulary to Faroese. Put simply, we observe slight gains over the ScandiBERT-fc3 model. We hypothesize that this might be due to the fact that including Faroese in the vocabulary results in a lower subword overlap with the other Scandinavian languages, which in consequence, slightly reduces the potential for transfer. While there is only a difference of 95 tokens between the two vocabularies, the difference yields 6% of the words in FC3 being tokenized differently.

Finally, the results also demonstrate the importance of focusing on a smaller set of related languages rather than relying on a broader set of languages from the MMTs. Unlike the results from Fujinuma et al. (2022), our results suggest that for languages with higher-resource 'siblings' such as Faroese, a higher-performing LM is a less general ScandiBERT model rather than an MMT such as XLM-R or mBERT. Different variants of ScandiBERT outperform XLM-R without any Faroese data across the board in all evaluation tasks. Another interesting finding is that additionally fine-tuning on Faroese data (the `-fc3` variants) has a much stronger positive impact on XLM-R as the underlying model than on ScandiBERT. Put simply, the importance of in-target language data decreases with the availability of more focused pretained LMs covering only languages related to the target language.

### 5.2 Additional Experiments

**Transfer with Wechsel.** To put our work in further context, beyond comparison to MMTs, we consider an alternative transfer learning approach, the Wechsel method (Minixhofer et al., 2022), a recent well-performing method for transferring monolingual Transformers to a new language. Further details and results are presented in Appendix B: they all show far worse performance than those presented

---

[3]We use the base-sized XLM-R: https://huggingface.co/xlm-roberta-base.

[4]Note that our study aims to establish how different pre-training strategies – and in particular languages included in pretraining – affect the models' downstream Faroese performance, rather than to investigate the downstream cross-lingual transfer. One could, naturally, additionally incorporate task-specific data in other Scandinavian languages (and also in English and other languages) in downstream training (i.e., perform cross-lingual transfer for the downstream task).

[5]Due to the limited size of the Faroese dataset, longer training with larger batch size consistently led to overfitting.

| Model | POS | | NER | | UD FP | | UD oft | | STS |
|---|---|---|---|---|---|---|---|---|---|
| | F1 | Acc. | F1 | Acc. | F1 | Acc. | F1 | Acc. | Acc. |
| IceBERT | 85.5 ± 0.19 | 85.2 ± 0.16 | 87.9 ± 0.54 | 96.4 ± 0.09 | 93.6 ± 0.06 | 94.6 ± 0.03 | 92.7 ± 0.32 | 94.2 ± 0.25 | 70.6 ± 1.9 |
| IceBERT-fc3 | 90.9 ± 0.06 | 90.4 ± 0.06 | 90.9 ± 0.41 | 98.9 ± 0.03 | 96.6 ± 0.06 | 97.1 ± 0.06 | 95.3 ± 0.38 | 96.1 ± 0.32 | 72.9 ± 1.8 |
| DanskBERT | 73.4 ± 0.19 | 74.3 ± 0.16 | 85.6 ± 0.44 | 98.4 ± 0.06 | 86.2 ± 0.16 | 87.7 ± 0.09 | 84.8 ± 0.57 | 88.7 ± 0.44 | 73.2 ± 1.3 |
| DanskBERT-fc3 | 87.1 ± 0.13 | 86.4 ± 0.13 | 89.7 ± 0.54 | 98.8 ± 0.06 | 96.0 ± 0.06 | 96.6 ± 0.03 | 94.2 ± 0.28 | 95.7 ± 0.19 | 75.3 ± 1.1 |
| XLM-R | 84.6 ± 0.28 | 85.0 ± 0.28 | 87.8 ± 0.47 | 96.3 ± 0.06 | 93.5 ± 0.06 | 94.3 ± 0.03 | 91.5 ± 0.44 | 93.6 ± 0.35 | 69.5 ± 2.1 |
| XLM-R-fc3 | 91.2 ± 0.09 | 91.2 ± 0.09 | 90.9 ± 0.41 | **98.9 ± 0.06** | 97.3 ± 0.06 | 97.7 ± 0.03 | 95.7 ± 0.22 | 96.8 ± 0.19 | 69.2 ± 2.1 |
| ScandiBERT-no-fo | 88.4 ± 0.09 | 88.1 ± 0.09 | 89.9 ± 0.25 | 96.7 ± 0.16 | 95.9 ± 0.06 | 96.4 ± 0.06 | 93.8 ± 0.35 | 95.0 ± 0.32 | 75.3 ± 1.5 |
| ScandiBERT-no-fo-fc3 | 91.5 ± 0.09 | 91.2 ± 0.09 | **91.4 ± 0.35** | 98.8 ± 0.06 | **97.4 ± 0.03** | **97.8 ± 0.03** | **96.3 ± 0.22** | **96.8 ± 0.19** | **76.5 ± 1.3** |
| ScandiBERT | 90.3 ± 0.09 | 90.0 ± 0.13 | 90.2 ± 0.28 | 99.0 ± 0.06 | 96.5 ± 0.06 | 97.1 ± 0.06 | 95.2 ± 0.32 | 96.2 ± 0.25 | 46.3 ± 6.3 |
| ScandiBERT-fc3 | **91.6 ± 0.06** | **91.3 ± 0.09** | 91.0 ± 0.35 | 99.0 ± 0.03 | 97.3 ± 0.06 | 97.7 ± 0.06 | 95.9 ± 0.25 | 96.7 ± 0.22 | 63.8 ± 6.2 |

Table 1: Results for all downstream tasks in Faroese using the different base language models, with and without continued Faroese pre-training. The -fc3 postfix indicates models that were further pretrained on FC3. Standard error intervals are also reported.

in Table 1. We hypothesize this is due to how closely related the languages we consider are, as opposed to the distant languages considered in the original Wechsel work.

**Task-Specific Transfer.** To explore the potential for task-specific transfer between closely related languages, we consider if labelled Scandinavian datasets can be combined to benefit Faroese. In particular, we look at NER as there is an easy way to map between labels of the different languages. See Appendix C for more details. The best result is achieved when training directly from the IceBERT model, which has been trained on the large MIM-GOLD-NER dataset, showing that given enough resources and a close enough language model, such a direct approach can be the most effective.

**Further Discussion.** Some of the results in Table 1 are as expected. Starting from the closest language relative, the Icelandic model, IceBERT, results in better performance for all downstream tasks than starting with the Danish model DanskBERT. The ScandiBERT model performs better than the massively multilingual XLM-R on all tasks, bar the more semantic FO-STS task.

What is more interesting is that the ScandiBERT-no-fo model that is not trained on Faroese outperforms the model that has Faroese included, when fine-tuned further on the FC3 dataset. In particular, for the higher level Fo-STS task. We hypothesize that this forces the Faroese adaptation to use the word segmentations from the related languages for a higher transfer benefit, as the tokenizing vocabulary was trained without Faroese. This is something we hope to investigate more in future work.

## 6 Conclusion and Future Work

We have shown that leveraging phylogenetic information and departing from the 'one-size-fits-all'

paradigm can improve cross-lingual transfer to low-resource languages. Our evaluation results show that we can substantially improve the transfer performance to Faroese by exploiting data and models of closely-related high-resource languages instead of relying on MMTs. In future work, we hope to extend the investigations and methodology beyond Faroese, to other low-resource languages for which higher-resource language relatives exist.

In order to boost and guide future research on Scandinavian languages in general and Faroese in particular, we make the models *ScandiBERT*[6], *ScandiBERT-no-fo*[7], *DanskBERT*[8] and *FoBERT (ScandiBERT-no-fo-fc3)*[9] available. As well as the new datasets *FC3*[10], *FoNE*[11], and *Fo-STS*[12].

## Acknowledgments

VS is supported by the Pioneer Centre for AI, DNRF grant number P1. The work of IV has been supported by a personal Royal Society University Research Fellowship (no 221137; 2022-).

We would like to thank Haukur Barri Símonarson for his comments on the work in its early stages. We also thank Prof. Dr.-Ing. Morris Riedel and his team for providing access to the DEEP cluster at Forschungszentrum Jülich.

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

## A  Training of language models

We train new BPE vocabularies for all the new models we train, ScandiBERT, ScandiBERT-no-fo, and

DanskBERT. All models use the same vocabulary size of 50k. The ScandiBERT vocabulary is trained using all the languages, the ScandiBERT-no-fo vocabulary is trained without the Faroese data, and the DanskBERT vocabulary is only trained on the Danish text. Vocabularies are trained using the SentencePiece software (Kudo and Richardson, 2018), and character coverage is set to 99.995 %.

Pre-training of the new language models is done using `fairseq` (Ott et al., 2019) using the RoBERTa-base (Liu et al., 2019) configuration, fine-tuning is done using the `transformers` (Wolf et al., 2020) library. ScandiBERT and ScandiBERT-no-fo were trained for 72 epochs, using a batch size of 8.8k sequences on 24 NVIDIA V100 cards for approximately 14 days each. Initial testing showed that the larger batch size showed better performance than going for around 2k sequences, possibly due to the mixture of differing languages. DanskBERT, on the other hand, similar to IceBERT and RoBERTa showed better performance at the smaller batch size. DanskBERT was trained to convergence for 500k steps using 16 V100 cards for approximately 14 days.

All *-fc* models are further trained for 50 epochs, with an effective batch size of 100k tokens for 12k updates, over the FC3 dataset for Faroese adaptation.

An overview of the data used to train the language models is shown in Table 2. For details on the Icelandic data, we refer to (Snæbjarnarson et al., 2022). For the other datasets, we refer to §2.

## B   Wechsel results

We compare our method to another transfer learning approach presented by Minixhofer et al. (2022). The FC3 dataset is used to train fastText embeddings for Faroese, and the Icelandic datasets are used to train fastText embeddings for Icelandic. These embeddings are then used to convert the multilingual models to Faroese using

| Language | Datasets | Size |
|---|---|---|
| Icelandic | IGC / IC3 / Skemman / Hirslan | 16 GB |
| Danish | Danish Gigaword Corpus (incl. Twitter) | 4,7 GB |
| Norwegian | NCC corpus | 42 GB |
| Swedish | Swedish Gigaword Corpus | 3,4 GB |
| Faroese | FC3 + Sosialurinn + Bible | 69 MB |

Table 2:   Datasets used to train ScandiBERT, ScandiBERT-no-fo and DanskBERT

the Wechsel approach. We confirm the quality of the Icelandic embeddings by running an Icelandic semantic evaluation suite adapted from `https://github.com/stofnun-arna-magnussonar/ordgreypingar_embeddings`, showing our embeddings are comparable or of higher quality than those released by Meta (Grave et al., 2018).

The experiments in Table 3 all show sub-par performance compared to the results in non-Wechsel results in Table 1. The Wechsel work considers transfer from English-dominant models, GPT2 and RoBERTa to French, German, Chinese, Swahili, Sundanese, Scottish Gaelic, Uyghur and Malagasy. None of which are closely related to English. One reason for the discrepancy in the results could be that the shuffling of the embedding matrix to convert it is more catastrophic when considering close languages. Another reason could be that both Faroese and Icelandic are morphologically rich and that all variants of the words were not properly mapped during the conversion of the embedding matrix.

## C   Mapping NER datasets

The datasets used to create a Scandinavian NER-corpus are DaNE (Danish), FoNE (Faroese), MIM-GOLD-NER (Icelandic), NorNE (Norwegian), and SWE-Nerc (Swedish), presented in §2. The results in Table 4 show that the best result is obtained when training directly from the IceBERT model. The ScandiBERT model has a higher variance when pre-fine-tuned on the combined NER corpora. This approach could also be made directly for the UD corpus, POS (in particular, using the Icelandic POS data), and other corpora as they become available for training or evaluation in Faroese. This demonstrates how resources from a related language can substantially benefit a low-resource language.

To combine the NER datasets, we map the tags to the CoNLL schema used by the Icelandic MIM-GOLD-NER and the Faroese FoNE datasets. The Danish DaNE dataset uses a subset of the tags used for Icelandic and Faroese, so the mapping is purely nominal. The mapping for Norwegian (NorNE) and Swedish (SweNERC) datasets is shown in Table 5.

## D   Prior Work on Transfer learning for Faroese

We know of three works that consider transfer learning for Faroese from the Scandinavian languages.

| | | POS | | NER | | UD FP | | UD oft | | STS |
|---|---|---|---|---|---|---|---|---|---|---|
| | Model | F1 | Acc. | F1 | Acc. | F1 | Acc. | F1 | Acc. | Acc. |
| Wechsel | IceBERT | 74.4 ± 0.16 | 75.7 ± 0.16 | 67.7 ± 1.2 | 98.7 ± 0.06 | 83.6 ± 0.35 | 84.6 ± 0.38 | 66.6 ± 9.01 | 75.7 ± 5.88 | 27.3 ± 3.9 |
| | IceBERT-fc3 | 89.0 ± 0.06 | 89.4 ± 0.09 | 88.5 ± 0.47 | 96.4 ± 0.09 | 96.3 ± 0.03 | 96.5 ± 0.03 | 95.6 ± 0.28 | 96.2 ± 0.25 | 67.7 ± 2.8 |
| | XLM-R | 68.9 ± 0.16 | 73.5 ± 0.13 | 59.7 ± 0.92 | 99.0 ± 0.06 | 81.0 ± 0.19 | 84.5 ± 0.13 | 71.8 ± 0.73 | 79.7 ± 0.44 | 11.4 ± 4.6 |
| | XLM-R-fc3 | 86.8 ± 0.09 | 88.7 ± 0.09 | 88.8 ± 0.41 | 98.4 ± 0.06 | 96.3 ± 0.03 | 96.7 ± 0.03 | 95.6 ± 0.25 | 96.5 ± 0.19 | 65.7 ± 2.8 |
| | ScandiBERT-no-fo | 71.3 ± 0.16 | 72.5 ± 0.16 | 65.1 ± 0.54 | 98.8 ± 0.03 | 82.0 ± 0.19 | 83.4 ± 0.19 | 75.0 ± 0.66 | 81.0 ± 0.57 | 29.4 ± 4.8 |
| | ScandiBERT-n.f.-fc3 | 89.2 ± 0.06 | 89.6 ± 0.06 | 89.2 ± 0.54 | 99.0 ± 0.03 | 96.8 ± 0.06 | 97.1 ± 0.06 | 96.1 ± 0.28 | 96.8 ± 0.22 | 74.7 ± 1.0 |
| | ScandiBERT | 72.6 ± 0.28 | 73.8 ± 0.28 | 65.7 ± 0.54 | 98.8 ± 0.03 | 83.0 ± 0.38 | 84.0 ± 0.28 | 76.8 ± 0.51 | 82.5 ± 0.35 | 8.7 ± 5.3 |
| | ScandiBERT-fc3 | 89.3 ± 0.09 | 89.7 ± 0.09 | 88.8 ± 0.54 | 98.7 ± 0.06 | 96.8 ± 0.03 | 97.1 ± 0.03 | 96.0 ± 0.25 | 96.7 ± 0.25 | 53.6 ± 6.0 |

Table 3: Results for all downstream tasks using different base language models after Wechsel adaptation, with and without continued Faroese pre-training. The results are significantly worse than without Wechsel adaptations.

| Model | Pre-ft. | Ft. | F1 | Acc. |
|---|---|---|---|---|
| SB-no-fo-fc3 | None | Yes | 91.4 ± 0.35 | 98.8 ± 0.06 |
| ScandiBERT | Icel. | Yes | **92.0 ± 0.32** | 98.8 ± 0.06 |
| ScandiBERT | All | No | 91.5 ± 0.51 | 98.9 ± 0.06 |
| ScandiBERT | All | Yes | 91.8 ± 0.51 | 99.0 ± 0.06 |
| XLM-R | All | No | 90.6 ± 0.19 | 99.0 ± 0.03 |
| XLM-R | All | Yes | 90.8 ± 0.47 | 99.0 ± 0.06 |

Table 4: NER performance when models are pre-finetuned on all Scandinavian datasets and then fine-tuned on FoNER.

| Language | Original | Mapped |
|---|---|---|
| Norwegian | O | O |
| Norwegian | PER | Person |
| Norwegian | ORG | Organization |
| Norwegian | GPE_LOC | Location |
| Norwegian | PROD | Miscellaneous |
| Norwegian | LOC | Location |
| Norwegian | GPE_ORG | Organization |
| Norwegian | DRV | O |
| Norwegian | EVT | Miscellaneous |
| Norwegian | MISC | Miscellaneous |
| Swedish | O | O |
| Swedish | EVN | Miscellaneous |
| Swedish | GRO | Organization |
| Swedish | LOC | Location |
| Swedish | MNT | Miscellaneous |
| Swedish | PRS | Person |
| Swedish | TME | Time |
| Swedish | WRK | Miscellaneous |
| Swedish | SMP | Miscellaneous |

Table 5: Mapping of tags to create a unified NER dataset for the Scandinavian languages.

In (Tyers et al., 2018), a rule-based translation system (Apertium (Forcada and Tyers, 2016)) is used to translate the Faroese Wikipedia into Swedish, Norwegian Bokmål, and Norwegian Nynorsk. The translations are then aligned, and the translations dependency-parsed. The resulting trees are then mapped to the original Faroese sentences and used for POS-tagging and annotating morphological features. The second work is a mapping between Faroese and Icelandic POS-tags (Hafsteinsson and Ingason, 2021); while not a direct application, the authors suggest the mapping may be of use for transfer learning between the languages. Finally, (Barry et al., 2019) use machine translation and dependency parsing for cross-lingual syntactic knowledge transfer from Danish, Norwegian, and Swedish to Faroese.