# OpenReview forum: "Transfer to a Low-Resource Language via Close Relatives: The Case Study on Faroese"
_NoDaLiDa/2023/Conference — NoDaLiDa 2023_

### Official Review · Reviewer_qiQB · 2023-03-09
**Interesting findings with room for some minor presentation improvements**

**Rating:** 8
**Confidence:** 4

**Review:**

This paper evalautes the effect of training language models on different data sources for downstream performance on a low-resource language (i.e. Faroese) in a high-resource family. The paper trains language models on Danish, Icelandic, Norwegian, Swedish and Faroese. They introduce a new filtered raw dataset (fc3)and two new evaluation datasets for Faroese (and report scores on other Faroese benchmarks). Their results show that training on all languages (including Faroese, especially fc3) is beneficial over single language language models, as well as massively multilingual language models.  Models and datasets will be shared.

The experiments are interesting, and the content is a good fit for a short paper. I had to go back and forth a bit to fully understand the setup (see comments below). But I particularly liked the variety of tasks and the comparison to Wechsel.

An overview table with the data sources used would make it much easier to understand the setup. It is for example important to know the sizes of LM data (appendix A), but also for the downstream tasks training data. Some space for this can easily be obtained by making section 1 and 2 smaller (the own contributions only start in the middle of page 3, more than 50% is thus intro+related work)

Clarify the setup earlier; You fine-tune on Faroese data in your main experiments, and also attempt to transfer from other languages for the NER task (this only becomes clear after seeing the results now). Also the motivation as to why you transfer for NER is not convincing, UPOS has the exact same labels already.

Smaller notes:
- Rename UD to UPOS in Table 1.
- missing reference: https://aclanthology.org/D19-6118.pdf
- it is unclear what no-fo is removing from the main-text, in appendix A I found that it is bible+ Sosialurinn.
- I would love to see a small qualitative analysis on the wechsel allignments
- 350: The overlap can quite easily be calculated, right?

**Paper Type:**

Short paper

---

### Official Review · Reviewer_UJMR · 2023-03-12
**A descirption of a method of transfer learning of Faroese language model using data and models of closely related language**

**Rating:** 5
**Confidence:** 4

**Review:**

The paper describes and compares two methods (as well some others) of traning Faroese language models based on the resources from closely related languages. In the first case, Faroese data is added to the traning data and multi-lingual language models are learned and in the second case mono-lingual or multi-lingual language models are trained first and then these are fine tuned/transfered to the Faroese data.

The paper tackels a very interesting topic and presents several relevant ideas, experiments and promising results but these are left with a lot of questions unanswered either because further expeirments are pending or there has not been enough room in a 4 pages to describe them. In fact, a lot of results and discussion in the appendix could be (with some further discussion) integrated with the main text. Overall, it is a highly promising paper but it does feel like work in progress and therefore would recommned it to be more suited for a workshop than a main ACL-style conference in its present form.

The authors describe a method of phyllogenic transfer but besides common knowledge about what languages are related do not use any specific phyllogenetic or linguistic information when training these models, e.g. language similarity scores, etc. which has been explored in the literature. It is also to expect the similarity between languages to be different in terms of syntax, vocabulary, discourse structure, etc. and such information is sometimes also exploited or shown how it affects performance on different tasks.

l.264 and following could give more details about the linguistic situation of Faroese. The phrasing "explicit upper bound to the amount available due to the small size of the Faroese-speaking population" is unfortunate - there is technically no upper-bound on language production - Faroese spekares only need to speak more obver longer time to generate the same amount of data :-) - and should be rephrased.

The difference between ScandiBERT-no-fo-fc3 and ScandiBERT-fc3 or the closest single-language BERT IceBERT-fc3 are very small and therefore the answer to the main research question of the paper is not very strong and conclusive.

l.418 "We have shown that by leveraging phylogenetic informationand departing from the ‘one-size-fits-all’ paradigm, one can improve cross-lingual transfer to low-resource languages." The paper has not shown this as it has not systematically compared the effect of language similarity on performance. Also, a model trained with other languages XLM-R-fc3 shows similar performance.

The work aligns with work on building resources in other under-resources scenarios for other languages. Due to a similiar linguistic situation work on dialectal Arabic is relevant here where similiar methods and approaches have been tried

W. Adouane. Natural Language Processing for Low-resourced Code-switched Colloquial Languages – The Case of Algerian Language. Doctoral thesis, Department of Philosophy, Linguistics and Theory of Science, University of Gothenburg, Gothenburg, Sweden, September 2 2020. http://hdl.handle.net/2077/64548

K. Abu Kwaik. Resources and Applications for Dialectal Arabic: the Case of Levantine. Doctoral thesis, Department of Philosophy, Linguistics and Theory of Science, University of Gothenburg, Gothenburg, Sweden, May 25 2022. https://hdl.handle.net/2077/71096

Typos:

l. 350, missing . and space between sentences, probably a result of editing. A couple of more such typos in the paper.

**Paper Type:**

Short paper

---

### Official Review · Reviewer_mEfw · 2023-03-14
**Important Datasets, Relatively Obvious Research Question**

**Rating:** 7
**Confidence:** 4

**Review:**

This paper is based on the assumption that it's generally standard to use highly multilingual pretrained models for crosslingual transfer. It them explores if using models pretrained on only related languages are better for a low-resource target language, Faroese. The problem I have with this question is that it's by no means new; it is a known result that transfer from related languages works better. Multilingual models are just used because it's easier, they can be taken out-of-the-box for a large set of languages. (For a larger analysis, please take a look at, for instance, this paper: https://aclanthology.org/2022.acl-long.106.pdf)

However, the paper is clearly written and the experiments seem rigorous. In addition, the authors present a couple of new datasets for Faroese, which will be important for research on that languages. Thus, overall, I do believe the paper should get accepted.

**Paper Type:**

Long paper

---

### Decision · Program_Chairs · 2023-03-17

Accept